# How Australian Rural Health Academic Centres Contribute to Developing the Health Workforce to Improve Indigenous Health: A Focused Narrative Review

**DOI:** 10.3390/healthcare13151888

**Published:** 2025-08-01

**Authors:** Emma V. Taylor, Lisa Hall, Ha Hoang, Annette McVicar, Charmaine Green, Bahram Sangelaji, Carrie Lethborg, Sandra C. Thompson

**Affiliations:** 1Western Australian Centre for Rural Health (WACRH), University of Western Australia, P.O. Box 109, Geraldton, WA 6531, Australiasandra.thompson@uwa.edu.au (S.C.T.); 2Rural Health Bendigo, Monash University, 26 Mercy Street, Bendigo, VIC 3552, Australia; 3Centre for Rural Health, School of Health Sciences, University of Tasmania, Locked Bag 1322, Launceston, TAS 7250, Australia; 4Southern Queensland Rural Health, University of Queensland, Boyce Ave, Cranley, QLD 4350, Australia; 5St Vincent’s Hospital, 41 Victoria Parade, Fitzroy, VIC 3065, Australia

**Keywords:** Indigenous, Aboriginal and Torres Strait Islander, First Nations, rural health academic centre, rural health workforce, health education, higher education

## Abstract

**Background/Objectives:** Improving health outcomes for Indigenous people by strengthening the cultural safety of care is a vital challenge for the health sector. University Departments of Rural Health (UDRH), academic centres based in regional, rural, and remote (RRR) locations across Australia, are uniquely positioned to foster a culturally safe rural health workforce through training, education, and engagement with Indigenous communities. This narrative review examines the contributions of UDRHs to health workforce issues through analysis of their publications focused on Indigenous health. **Methods:** Research articles relating to workforce were identified from an established database of UDRH Indigenous health-related publications published 2010–2021. **Results:** Of 46 articles identified across the 12 years, 19 focused on developing the understanding and cultural safety skills of university students studying in a health field, including campus-based Indigenous health education and support for students undertaking rural clinical placements. Twelve articles investigated cultural safety skills and recruitment and retention of the rural health workforce. Fifteen articles focused on Indigenous people in the health workforce, examining clinical training and resources, and the enablers and barriers to retaining Indigenous students and workers. **Conclusions:** This analysis highlights the sustained efforts of UDRHs to improve Indigenous health through multiple areas within their influence, including curriculum design, health student training on campus, and rural placement opportunities to transform understanding of Indigenous strengths and disadvantages and rural health workforce development. A continuing effort is needed on ways UDRHs can support Indigenous health students during their studies and while on placement, how to improve cultural safety in the health workforce, and ways to better support Indigenous health professionals.

## 1. Introduction

Aboriginal and Torres Strait Islander peoples are the original inhabitants of Australia and experience large, persistent health disparities, with shorter life expectancy and higher rates of chronic diseases compared to non-Indigenous Australians [1]. (Hereafter, the term ‘Indigenous Australians’ is respectfully used to refer to Australia’s Aboriginal and Torres Strait Islander peoples, while acknowledging the diversity of cultures and experiences of Australia’s First Peoples.) These disparities result from the continuing impacts of colonisation with lasting effects on determinants of health and access to appropriate healthcare [2]. Barriers to healthcare include fear or mistrust of mainstream healthcare (due to past negative experiences or fear of negative government interventions) and poor communication, disrespect and racism from health professionals, with experiences of racial bias affecting patients’ willingness to engage with care and leading to misdiagnosis and poorer outcomes [3,4,5]. The 59% of Indigenous Australians living outside of major cities experience additional barriers to accessing healthcare, including poorer continuity of care, lack of access to specialist care and costs of travelling to access healthcare [6,7,8].

The accessibility and cultural safety of health services and health staff significantly impact on Indigenous people accessing those services, with multiple studies reporting the importance of Indigenous staff involved in healthcare [3,9,10]. The National Aboriginal and Torres Strait Islander Health Plan 2021–2031 states that “cultural safety is about how care is provided, rather than what care is provided. It requires practitioners to deliver safe, accessible and responsive healthcare that is free of racism by recognising and responding to the power imbalance between practitioner and patient; and reflecting on their knowledge, skills, attitudes, practising behaviours, and conscious and unconscious biases” [11]. Training, recruitment, development, and retention of a culturally safe health workforce are all critical for meeting the healthcare needs of Indigenous Australians. Ways to develop a culturally safe workforce include cultural safety education and culturally immersed clinical placements for health students, developing the cultural understanding and skills of the health workforce and supporting the development and expansion of the Indigenous health workforce [12,13,14]. However, multiple challenges for rural and remote health services in attracting and retaining a culturally safe workforce are documented [15,16].

University Departments of Rural Health (UDRH) are academic centres based in regional, rural and remote (RRR) locations across Australia, and are funded by the Australian Government Department of Health, Disability and Ageing through the Rural Health Multidisciplinary Training Program (RHMT) [17]. The Australian Government has invested in UDRHs for over 25 years as part of its solution to addressing the health workforce shortage that occurs in rural Australia. UDRHs facilitate RRR training and placements for nursing and allied health students, support the continuing professional development of rural health professionals, and undertake research on rural and remote health issues and so are uniquely positioned to help build a culturally safe rural health workforce [18]. UDRHs generally facilitate cultural awareness opportunities for students on placement and may partner with local Indigenous communities, provide mentoring to Indigenous health students, support the local Indigenous health workforce and undertake locally relevant research. Research focus areas include rural and remote health workforce development and improving the health and wellbeing of Indigenous Australians. The Australian Rural Health and Education Network (ARHEN) is the national association for the 19 UDRHs, providing leadership and strategic direction, as well as facilitating UDRHs to share information and collaborate to improve the health and wellbeing of rural communities [17]. In 2019 ARHEN established an Endnote database to serve as a central repository of UDRH, training and workforce research from 2010 onwards.

This study was conducted as part of a broader investigation to identify and describe the contributions of UDRHs to Indigenous issues through analysis of the ARHEN Endnote database during the period 2010–2021. The previous study explored the number and type of publications with a focus on Indigenous health that were published by the UDRHs between 2010 and 2021 [19], and as foreshadowed in that article, there was an intention to explore in more detail some of the research categories. Interestingly, despite the focus of UDRHs on building the rural health workforce and supporting Indigenous health, the number of articles (45/493) that specifically address workforce issues related to Indigenous health was surprisingly small, and we call for further attention in this area [19]. The aim of this review is to examine the workforce articles with a focus on Indigenous health, as identified in our previous study [19], with the intention of identifying the contributions of UDRHs to key learnings relevant to improving Indigenous health and gaps in the research.

## 2. Methods

A research team of eight people from four UDRHs was formed in October 2023. The research team consisted of two Indigenous researchers; AM, a Mununjahli and Minjungbal woman with a background in education and training and CG, a Wajarri, Badimaya and Wilunyu woman and experienced researcher. The non-Indigenous members of the research team have extensive experience in Indigenous health research, cultural education and training, and rural health.

### 2.1. Identification and Screening

This narrative review analysed Indigenous health workforce papers that were identified and classified as workforce papers in our previous study [19]. Articles that specifically examined Indigenous issues related to the health workforce published between 2010 and 2021, and with at least one UDRH-affiliated author were included. Our previous study identified 493 UDRH publications in Indigenous health sourced from the ARHEN Endnote database, and coded articles into the following broad categories: health services research; epidemiology, Indigenous culture and needs for health and wellbeing; workforce issues; and other. Details of the search strategy and screening to identify the Indigenous health research articles can be found in our previous publication [19]. The previous study identified 45 research publications on the workforce and recommended exploring those publications in greater detail, ref. [19] which therefore form the subject of this narrative review.

Articles that considered workforce issues such as training the future health workforce, recruitment, retention, and support of the existing workforce, and professional education around the care of Indigenous people were classified as workforce articles, with three subcategories identified:Indigenous health workforce: supporting and developing Indigenous Australians in the health workforce, including issues related to recruitment and retention.Health workforce: developing cultural safety/skills in health professionals and issues related to recruitment and retention.Students: developing cultural understanding, safety, or skills in health students studying at university.

To ensure that no relevant articles were missed, the original set of 493 publications from the previous study was re-examined by the research team, and six additional articles were identified and included, bringing the total for this study to 51. In the original breakdown, systematic reviews were reported separately to research articles; however, as the aim of this review was to consider UDRH research contributions, three workforce reviews were deemed relevant and included. In addition, three articles originally categorised as “health services research” or “epidemiology” were identified as relevant to workforce issues and included. However, during the initial screening five articles were considered not sufficiently relevant to workforce and were excluded. Article inclusion and exclusion criteria are shown (Table 1). Results for each stage of the identification and screening processes are shown (Figure 1).

### 2.2. Data Extraction

Five authors (E.V.T., L.H., H.H., A.M. and S.C.T.) were each allocated a subset of the 46 papers to screen and review, with all authors reviewing approximately nine articles. Each author extracted information on the key findings and recommendations of their assigned articles as relating to the student workforce, health workforce and Indigenous Australian health workforce using a data extraction template created in Microsoft Excel. The extracted information was cross-checked by at least one additional author to ensure accuracy. Any discrepancies were resolved through discussion within the wider research team. Articles were also coded for authorship affiliation, the state where the research was conducted, and the geographic remoteness of the research using the Australian Statistical Geography Standard (ASGS) Edition 3 [20].

### 2.3. Data Analysis

Due to the heterogeneous nature of the articles across the three sub-categories, articles within each sub-category were coded independently by at least two authors for themes using NVivo 13. Coding was an iterative process, with articles in each category first grouped into one or more overarching themes. Once the themes were agreed, team discussion occurred to reach consensus on the main themes with refinement/amalgamation of categories and identification of subthemes. Any papers that presented challenges were resolved by discussion with at least one additional reviewer. Excel was used for descriptive analysis and reporting simple frequencies and percentages.

## 3. Results

### 3.1. Characteristics of the Papers Reviewed

There were 46 workforce-focussed articles across the 12-year period, with the number of papers substantially increasing, from 14 in 2010–2015 to 32 in 2016–2021 (Figure 2). Appendix A displays a list of the 46 publications and a summary of key findings and recommendations.

Almost half the articles reported research using qualitative methods (n = 22, 48%) (Table 2). Most were co-authored by UDRH authors in collaboration with authors from university departments and industry partners (n = 33, 72%). Sixteen articles reported on research occurring across multiple levels of remoteness (34%), 24 reported on regional, remote or very remote areas (52%), and three, all reporting on the training of health students about Indigenous culture and health, reported research that took place on urban campuses.

### 3.2. Student Workforce Publications

Working with students studying in a health field around Indigenous health and culture was the largest category, comprising 19 articles. These articles covered both campus-based Indigenous health education and support for students undertaking rural clinical placements and cultural immersion experiences (Table 3). Most participants were non-Indigenous students from medical, nursing, midwifery, and allied health disciplines. Only one study, a comprehensive review of issues related to Indigenous student retention in health sciences courses, solely focusing on Indigenous students [21].

#### 3.2.1. Campus-Based Education

Campus-based Indigenous health education revealed diverse perspectives and challenges [21,22,23,24,25,26,27]. A review found that Indigenous health education focused on remembering and understanding with limited emphasis on application, cultural safety, and security [22]. While most students received education on Indigenous health, many still felt underprepared for practice [23]. Factors contributing to preparedness included previous educational experiences, clinical exposure, and teaching by Indigenous educators [23]. Ensuring that campus-based educators were themselves prepared and had embraced understanding of Indigenous knowledges was critical, with the involvement of Indigenous staff as part of the delivery of the education essential for this [21].

The completion of an Indigenous Health and Wellbeing unit did not significantly impact students’ opinions on cultural desire, despite increasing understanding of Indigenous health [24]. Cultural desire is a term coined by Campinha-Bacote to refer to motivation and genuine desire for cultural understanding, as opposed to the obligation of encountering cultural diversity [40]. Students articulated the need to be in a safe and non-judgemental learning environment to discuss challenging aspects of Indigenous experiences of health service provision, for example, issues around racism [26,28]. Receptivity of students to such discussions varied, with some students upset that they had not previously learned the extent of oppression experienced by Indigenous Australians, while others were uncomfortable and made comments such as “issues were made to be Aboriginal issues when they could also be applied to anyone” ([25], p. 17). While stereotypes were challenged, issues related to racism remained unresolved, with some students dismayed by the attitudes of their peers.

The only study focussed solely on Indigenous students found that factors affecting their retention included family support, academic preparation, and experiences of racism [21].

#### 3.2.2. Clinical Placement and Immersion Experiences

Several papers highlighted the impact of immersion experiences on individuals and on the community [30,31,32,33,34]. Experiential learning in small rural and remote communities helped students to appreciate how much could be learnt from Indigenous people and understand the impact of social and cultural determinants of health on Indigenous Australians. Clinical placements in rural areas positively influence attitudes, preparedness for practice, and understanding of rural communities and health issues [35]. Remote clinical placements were valued for offering profound learning experiences, building connections with community members and the opportunity to apply insights around Indigenous cultural ways to clinical and community practice [31,32]. Thackrah et al. (2015) stressed the vital role played by culturally respectful approaches, the informal provision of culturally relevant information, and the involvement of Indigenous Health Workers and community members in care delivery which students became aware of through their exposure in a rural placement [33]. Factors contributing to positive learning experiences included pre-placement cultural training, peer support, community engagement, and relationship-building within the community [31].

#### 3.2.3. Integrating Indigenous Culture into Healthcare Education

Central to both campus-based and placement/immersion experiences was a focus on Indigenous health outcomes, the need to meet and learn from Indigenous Australians, preparing to join the health workforce, increasing self-knowledge and changing perceptions. Educating students on the causes of health disparities for Indigenous people was essential as this understanding helps build context around why these disparities exist [23,30]. An environment that incorporates Indigenous knowledge and practices, along with research grounded in Indigenous wisdom, can lead to improved methods of educating future healthcare providers [23]. Comprehensive cultural training before practical placements, focusing on building rapport and developing cultural understanding, was found to be vital [30].

Within this collection of articles, Thackrah and colleagues conducted extensive research on Indigenous health education for mostly non-Indigenous midwifery students [25,26,27,31,32,33,36,38]. Their studies covered various aspects, including classroom learning and remote placements. Midwifery students on remote placements had considerable opportunity for interaction with community members through supervised health service provision. This resulted in a deeper understanding of the significance of cultural protocols, especially those surrounding “women’s business”, the importance of silence in communication, the cultural sensitivities around sexual health and the important role of grandmothers in providing appropriate guidance to both younger women in the community and the students [32,38]. Their research also highlighted the importance of pre-placement cultural training, community engagement, and interprofessional collaboration. Despite challenges related to supervision and workloads, longitudinal follow-up found these placements had a lasting impact on desire to engage with Indigenous communities and improve care for Indigenous people [36].

### 3.3. Health Workforce Publications

Twelve articles reported on issues related to the health workforce, issues of care delivery, and how these affected the health of Indigenous Australians. The greatest number of papers came from the Northern Territory (NT), and papers were overwhelmingly descriptive. The considerable diversity in this broad category of papers often centred around the need to improve the cultural understanding and ability of health service staff to deliver culturally secure care and understanding the extent and impact of workforce turnover (Table 4).

Several papers reported on inadequate cultural knowledge in the health workforce, and how this lack of cultural capacity had a detrimental impact on Indigenous health outcomes with patients disengaging with healthcare and with poor communication leading to misdiagnosis [41,42,43,44,45]. Clinical skills were essential for the Remote Area Nursing workforce, but a lack of cultural skills resulted in Indigenous clients not returning for follow up, and there were consequences related to communication and misunderstanding [42]. A mixed methods study highlighted a considerable gap between the perceptions of primary healthcare providers and Indigenous community members on the provision of culturally appropriate primary healthcare services, particularly around issues of communication and lack of respect shown by reception staff [45]. The research highlighted the need for cultural awareness training to be developed with local Indigenous community leaders, to run throughout the year, and to include non-clinical staff. Several other articles commented on the effectiveness of cultural awareness training [13,41,42,44], and that such training needed to be of at least one full day’s duration [41].

The importance of Indigenous health professionals in providing culturally safe health services for Indigenous Australians and in improving the cultural knowledge of their non-Indigenous colleagues was emphasised by several articles [13,41,42,45,46]. There was a need for greater valuing of Indigenous Health Workers as cultural brokers, and for providers to actively involve them in the healthcare team [45]. Only one article talked about the negative impact of racism and explored the differences between non-Indigenous and Indigenous nurses’ perceptions of and responses to racism in the workplace [13].

Various methodological approaches were used but it was clear that workforce turnover impacted on quality of care, particularly in community settings and primary care. The very high turnover rates of health professionals working in remote clinics in the NT was described in four papers [16,42,48,49], with turnover impacting the willingness of Indigenous patients to present to the health services [42]. The quantitative findings from Jones et al. (2021) analysis of NT data noted the variation in care across clinics, but their analysis did not find evidence that staff turnover and reliance of temporary staff negatively impacted quality of care [49]. This surprising finding may reflect that quantitative data on health system performance does not reflect the views of Indigenous people who receive the care. Suggested strategies to support the remote health workforce included changes to the roster, remote scholarships and incentives for working remotely [16,48], although these were not evaluated. Tyrrell et al. (2020) took a novel approach to remote health workforce turnover by developing a model for estimating likelihood that a health professional would remain working in very remote Indigenous communities for more than three years, while noting the tool required further validation [50].

### 3.4. Indigenous Australian Health Workforce Publications

Fifteen articles were identified relating to the Indigenous health workforce, including several that reported on interventions, although most were descriptive. This category covered culturally supportive clinical training and resources for the Indigenous heath workforce as well as enablers and barriers to supporting and retaining the Indigenous workforce; all articles were published in the last 10 years. The need for creating supportive environments, addressing cultural perspectives, and overcoming systemic challenges to enhance the wellbeing and professional advancement of Indigenous health professionals was highlighted across the fifteen articles (Table 5).

The importance of culturally appropriate and supportive resources created with Indigenous cultural considerations and customs for Indigenous health professionals was identified by several papers [51,52,54,55,56,57,58]. Active, continuing engagement and consultation with the Indigenous community and Indigenous health professionals was essential for establishing effective advocacy and creating culturally relevant resources [52,53,54,55,56,57]. However, “greater appreciation for Aboriginal peoples’ rich traditions of storytelling” was needed in the development of culturally appropriate health resources ([58], p. 59). A culturally appropriate medication safety programme for rural Indigenous health professionals successfully improved knowledge, confidence and behaviour [56]. The success of the programme, developed to address gaps identified by Indigenous health professionals, was attributed to the collaborative approach taken to develop the educational materials.

Access to training opportunities for career development was important for building the capacity and career satisfaction of Indigenous people in the health workforce. Engagement in authentic learning experiences in practical contexts can enhance curiosity and interest, facilitate exploration and increase skills and create a positive influence on Indigenous workers’ willingness to adopt new practices [51]. A study found supervised clinical placements were an effective way for Indigenous health professionals to develop their skills in palliative care, and placements could improve confidence, relationships, and partnerships, however management support was critical [57].

Recognising and addressing barriers such as negative workplace culture, difficulties balancing community and career expectations, lack of support networks, low salaries, and lack of career opportunities were crucial factors for retaining Indigenous health professionals [15,60]. A workplace that values Indigenous culture and offers access to clinical and cultural support is critical to job satisfaction and helping empower the Indigenous health workforce [12,15,59]. Boundary setting and juggling family, cultural, community obligations with workplace expectations were particularly challenging for Indigenous health professionals in rural communities and contributed to feelings of burnout and may negatively affect retention [60]. Therefore, supportive managers and workplaces may not only reduce work stress but also help combat burnout and ultimately improve retention [12,15,59]. To create culturally sensitive workplaces, cultural safety training for all health workers is required [12,15].

Another challenge to retention described was the way many Indigenous positions were funded [15], with low or short-term funding of positions, salaries perceived to be lower than other health professionals, and greater levels of unpaid overtime. This caused some Indigenous health professionals to move to higher-paying positions or leave the health sector entirely [60]. Retention could be improved for when workplaces provided long-term or permanent positions, appropriate and fair remuneration, recognised cultural expertise as well as clinical expertise and offered access to flexible working arrangements [12,15,59].

Culturally appropriate clinical supervision for Indigenous health workers was found to be vital [57,61] with competent supervisors experienced in working with Indigenous people essential for effective support. Indigenous mentors as a means of providing culturally safe support for the Indigenous health workers was also deemed vital [59,61].

Systemic issues such as low expectations, colourism-based discrimination, and not having qualifications recognised were barriers to career progression [60,62,63]. Plater and colleagues 2020 reported that Indigenous professionals, including health professionals, experienced discrimination based on skin shade (known as colourism and referred to as shadeism in the article), where darker skin appeared to be associated with lower expectations of capability by non-Indigenous colleagues and managers [62]. Conversely, Plater also found that “dark-skinned people may be perceived as more racially and culturally authentic than light-skinned people and may therefore be considered desirable additions to workplaces where Aboriginal and Torres Strait Islander presence is a requirement” ([62], p. 494). Several studies reported participants feeling like their qualifications were not recognised by employers and having limited career opportunities compared to non-Indigenous health professionals [60,62]. Logistical barriers to career progression, including geographical constraints, telecommunication coverage, and financial limitations were also highlighted [63].

## 4. Discussion

This analysis highlights the contribution of UDRHs to efforts to improve Indigenous health, including through health student training and a better prepared health workforce. It brings together insights on supporting and developing Indigenous Australians in the health sector. The papers are diverse and demonstrate the efforts made in many UDRHs to engage with local Indigenous people and communities as part of addressing their health concerns.

The total number of UDRH publications over the twelve-year period was 4328, with 493 (11%) primarily focused on Indigenous issues. A previous review of UDRH publications identified 415 papers related to workforce [64] covering multiple areas including: health student placements, student and workforce education and rural health workforce characteristics, recruitment, retention and interventions. The review did not focus on Indigenous issues but noted many publications focused on Indigenous health, wellbeing, and care.

The 46 articles related to both workforce and Indigenous people identified in this study highlight the UDRHs interest in this area. This is consistent with their key role in workforce development and longstanding commitment to improving Indigenous health. The articles covered three broad categories: preparing health students to be culturally safe health professionals; skilling the existing health workforce to better understand Indigenous health and culture; and supporting more Indigenous Australians to enter or stay in the health workforce, including through safer university and work environments.

UDRHs have worked with communities, universities and health providers to improve the health and wellbeing of Indigenous people. Much of their work has focused on better preparing students to enter the health workforce, to appreciate Indigenous peoples and their rich cultures and to learn about culturally safe practice in the context of historical maltreatment and ongoing trauma and social disadvantage. This included evaluating curriculum efforts to expose students to Indigenous health and culture, while recognising that knowledge is not understanding and may not influence practise. Student learning was enhanced in small remote communities where students had opportunities for deeper engagement and to build relationships with community. A recent UDRH study found that longer placement length was associated with increased confidence working with Indigenous people [65]. Cultural immersion and remote learning programmes in small communities were found to have a profound impact on students, with longitudinal follow up showing that the legacy of such placements can be enduring [13,36], including greater understanding of Indigenous people and health, and increased commitment to rural and remote healthcare. These findings have influenced how UDRHs deliver their programmes, collaborating closely with Indigenous personnel and communities to train the future health workforce.

For many years, Indigenous Australians have reported a lack of cultural security in mainstream providers and health services, which impacts their willingness to use care [66,67]. Given that culturally safe practice is a requirement of registration by the Australian Health Practitioner Regulation Agency (AHPRA) and accrediting bodies of health disciplines, UDRHs are actively working to improve cultural safety in health services and the rural workforce. This has included efforts to make health services more attentive to the needs of Indigenous Australians, including better communication and models of care across urban and rural services. Many publications were related to remote areas where cultural dissonance may be greatest, and efforts to deliver high quality, culturally safe care require even more thought. These issues were important for considering recruitment, retention and turnover, the challenges experienced by the health workforce and relevant support strategies.

Developing the Indigenous health workforce is a critical issue across Australia with numerous studies identifying the need to include Indigenous health staff in the provision of care for Indigenous people [9,68,69]. Over the 12-year period, only 15 UDRH papers have focused on this issue. These studies emphasised the importance of flexibility, Indigenous leadership, peer support and understanding of Indigenous people and culture, and the need for relationships, partnerships, and culturally appropriate approaches and resources, informed by community input. The systematic reviews of Taylor et al. (2019) and Lai et al. (2018) pulled together work from the broader literature to, respectively, highlight the important elements for supporting Indigenous health science students in their journey through university, and how best to support Indigenous health professionals in the health workforce [15,21]. Several UDRH papers reported on interventions working with Indigenous people directly to improve Indigenous health outcomes: mental health [51,52,54], Indigenous men’s health [58] and medication safety [56]. This is a critical area demanding more attention: to develop the interest of Indigenous people in health careers, to support their learning in health courses and encourage translation of existing knowledge into practice through support for services and employers around better practices for recruitment and retention of Indigenous staff.

### Limitations

This analysis relied upon UDRHs having submitted their publications to the Australian Rural Health Education Network (ARHEN) so possibly not all publications were captured. Coding articles requires judgement, and with more in-depth analysis several articles were reclassified to other categories. We did not code for Indigenous authorship over the period under consideration, as there was no way to reliably identify the Indigenous status of authors. Moreover, Indigenous identity is complex [70] and has been changing over the time period in question [71], as have standards for the conduct of Indigenous health research. We do know that many of the UDRHs and authors have worked closely with Indigenous people and academics, employ Indigenous staff as co-researchers and include them as authors, and we know that considerable development in skills and knowledge occurred in both Indigenous and non-Indigenous people through the research and two-way learning.

A collaborative and capacity building approach was taken in conducting this research, with one member of the team new to conducting reviews. This inclusive process resulted in the analysis and write up taking longer, consequently relevant UDRH articles have undoubtedly been published since the articles were selected and the analysis was conducted.

## 5. Conclusions

UDRHs have worked with communities, health services, and universities in multiple ways to improve the health and wellbeing of Indigenous people. A considerable proportion of this work has focused on better preparing students to enter the health workforce, to appreciate Indigenous people and their rich cultures, and to learn about culturally safe practice in the context of historical maltreatment and ongoing trauma and social disadvantage. The rural and remote location of UDRHs opens the possibility for expediting student learning about Indigenous people, culture, and health. While classroom efforts offer a foundation for knowledge, the empathy and understanding gained by working closely with Indigenous people in more remote locations highlights the special contribution that UDRHs can make to developing the culturally safe workforce required by AHPRA. Efforts have also been made by UDRHs to improve the understanding of the existing workforce and to develop and support the existing Indigenous health workforce.

Future efforts should focus on encouraging local Indigenous people into health careers and evaluating the ways UDRHs support Indigenous health students during their studies and while on placement. Due to the small numbers of students, this may require UDRHs to work in partnership across jurisdictions and state borders. Any such research should be performed by engaging Indigenous researchers and educators, as well as local communities. UDRHs are uniquely situated to partner with rural and remote health organisations to improve cultural safety in health services and the health workforce to develop better ways to support Indigenous health professionals and thereby contribute to lasting advancements in Indigenous health and wellbeing.

## Figures and Tables

**Figure 1 healthcare-13-01888-f001:**
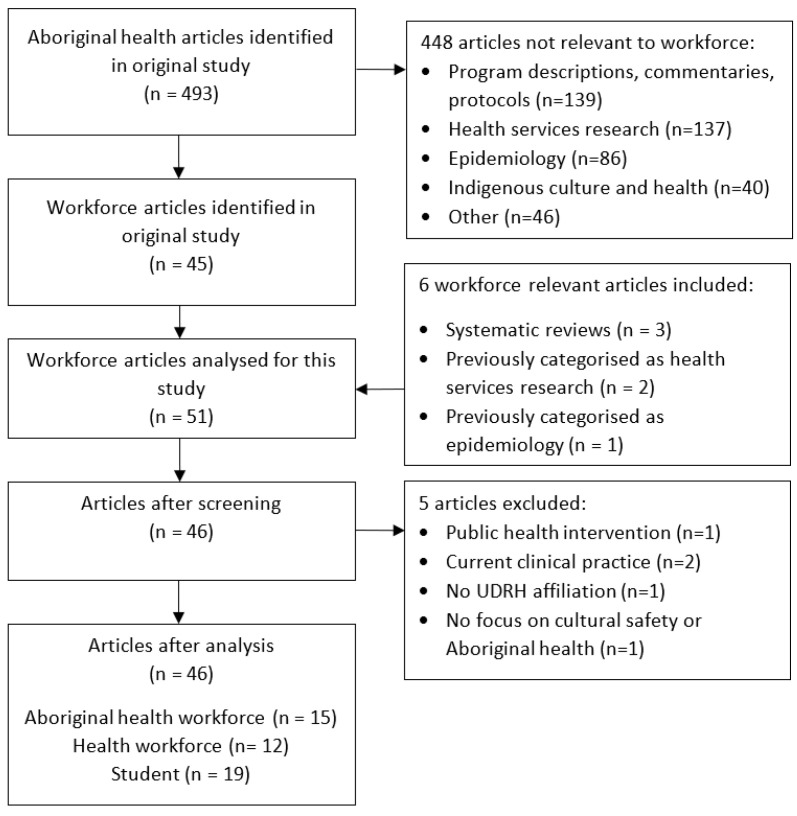
Flow diagram representing article identification and screening process.

**Figure 2 healthcare-13-01888-f002:**
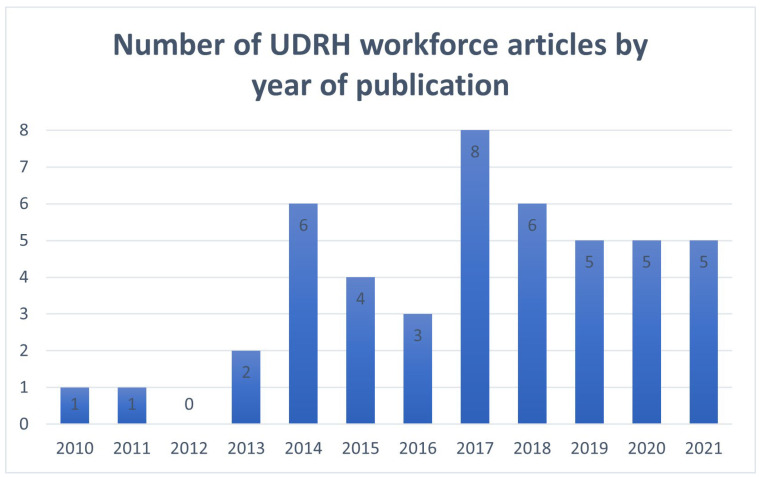
Number of UDRH workforce articles by year of publication.

**Table 1 healthcare-13-01888-t001:** Article Inclusion and Exclusion Criteria.

Domain	Included	Excluded
Time Period	2010 to 2021	Prior to 2010 and post 2021
Language	English	Non-English
Journal	Refereed (peer reviewed) journals only	Grey literature, conference proceedings, published abstracts
Type of articles	Research articles and reviews	Opinion pieces, commentaries and editorials, letters that are not research related
Scope	Focus on workforce issues such as training the future health workforce, recruitment, retention and support of the existing workforce, and professional education around the care of Indige-nous people	Not workforce related, or describing a health intervention or current clinical practice
Article authorship	At least one author’s organisational affiliation listed as a UDRH	No UDRH affiliation listed
Setting/location	Australia	Studies conducted outside of Australia

**Table 2 healthcare-13-01888-t002:** Number of UDRH workforce publications: methodology, authorship affiliation and location.

**Research methodology**	**Publications**
Qualitative	22
Quantitative	12
Mixed Methods	9
Systematic Review	3
**Authorship affiliation**	**Publications**
UDRH in partnership with University, Industry or ACCHO	33
UDRH alone	10
Co-authors affiliation unclear	3
**Location remoteness**	**Publications**
Multiple levels of remoteness	16
Remote or very remote	13
Inner or outer regional	11
Major city	3
**Location by state**	**Publications**
Research conducted across multiple states	12
Western Australia	10
New South Wales	7
Victoria	6
Northern Territory	5
Queensland	3
Not applicable	3

**Table 3 healthcare-13-01888-t003:** Student workforce publications by themes and sub-themes.

Themes	Sub-Themes Including Publications Where Data Were Retrieved
Campus-based Indigenous health education	Indigenous health curriculum [21,22,23,24,25,26,27]
	Cultural awareness [22,28,29]
	Racism [21,25,26]
	Learner reactions [22,24,25]
	Student retention [21]
Clinical placement and immersion experiences	Impact of placement (on individuals and on the community) [30,31,32,33,34]
	Building rapport and relationships [30,31,32,35]
	Transferability of knowledge and skills [32,36]
	Reflective practice [37]
Integrating Indigenous culture into healthcare education	Indigenous health disparities and outcomes [22,23,24,28,30,35,37]
	Learning from Indigenous people [23,26,27,28,30,31,32,33,34,38]
	Culturally relevant care [30,31,33,34]
	Preparation for workforce [22,23,26,28,30,31,34,37,39]
	Increasing self-knowledge and changing perceptions [22,25,26,27,28,34,35,36,38]
	Cultural safety and security for Indigenous people [22,23,26,28,37,38]
	Pre-placement learning [30,31]
	Cultural competence [24,29,30]
	Working rurally [31,32,34,35,36,37,39]
	Experiential learning [27,32,35,36,39]

**Table 4 healthcare-13-01888-t004:** Health workforce publications by themes and sub-themes.

Themes	Sub-Themes Including Publications Where Data Were Retrieved
Cultural safety of the health workforce	Lack of cultural knowledge [41,42,43,44,45]
	Cultural awareness training [13,41,42,44,45]
	Importance of Indigenous health professionals [13,41,42,45,46]
	Racism [13]
Remote workforce issues	Workforce over time [47,48]
	Turnover [16,42,48,49]
	Recruitment and Retention [42,50]
	Challenges [42]
	Clinical skills [42,43,47]
	Qualifications in remote health [47]
	Support strategies [16,48]
	Support for Indigenous health professionals [16,46]

**Table 5 healthcare-13-01888-t005:** Indigenous health workforce publications by themes and sub-themes.

Themes	Sub-Themes Including Publications Where Data Were Retrieved
Clinical training	Culturally appropriate resources [51,52,53,54,55,56,57,58]
	Training accessibility [51,57]
Workforce issues	Retention [12,15,53,59,60]
	Clinical supervision [57,61]
	Barriers to career progression [60,62,63]

## Data Availability

All data generated or analysed during this study are included in this published article.

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
