# Peer review of "How Australian Rural Health Academic Centres Contribute to Developing the Health Workforce to Improve Indigenous Health: A Focused Narrative Review"

_healthcare, 2025, doi:10.3390/healthcare13151888_

Round 1
Reviewer 1 Report
Comments and Suggestions for Authors
This manuscript addresses an important topic. Weaknesses of the manuscript are 1) the high self-citations, although this is partly due to the review nature of the manuscript; 2) that the search was limited to 2010-2021 and 3) limiting the search using only ARHEN. The manuscript is still valuable and I have made suggestions for improvements below.
- Clarification of UDRH Initiative
On page 2, it would be helpful to provide a brief explanation or context for the UDRH (University Departments of Rural Health) initiative. For readers unfamiliar with this framework, specifying the departments involved would enhance clarity and accessibility. - Geographic Representation
I was surprised to see that no studies from South Australian locations were included in the review. Could the authors clarify whether this was due to a lack of submissions from that region, or if there were other reasons for their exclusion? - Scope of Literature Search
The decision to limit the review to articles submitted to ARHEN warrants further explanation. Was this a methodological choice based on scope, or were there constraints that prevented a broader search? Clarifying this would strengthen the transparency of the review process.- Additionally, the literature search was limited to 2010-2021. Even a quick Google Scholar search shows several articles that could be included in this search. If at all possible, I would encourage the authors to extend their search to include more recent articles.
- Terminology: “First Nations” vs. “Indigenous”
I recommend that the authors consider using the term “First Nations” in place of “Indigenous,” as this is increasingly recognized as the preferred terminology in Australia. A brief note on terminology choices would be helpful for readers. - Self-Citation Transparency
Of the 46 papers reviewed, 23 include at least one author from the current manuscript. While self-citation is not inherently problematic in a review article, I encourage the authors to be more transparent about this overlap. For example:- On page 2, in the final paragraph, please clarify that the Thompson article was authored by the majority of the current team.
- On page 3, revise “was recommended” to “we recommended” to reflect authorship.
- Consider including a brief statement acknowledging the proportion of self-authored articles and describing any efforts made to identify and include work from other research teams to mitigate potential bias.
- Authorship and Indigeneity
On page 13, the authors note the difficulty in reliably coding for Indigenous authorship. However, the Indigeneity of the current authors is not made transparent. A brief statement on this would contribute to the integrity and reflexivity of the work. - Minor Correction
On page 7, please correct the spelling of “Campinha-Bacote.”
Author Response
Comment 1: Clarification of UDRH Initiative On page 2, it would be helpful to provide a brief explanation or context for the UDRH (University Departments of Rural Health) initiative. For readers unfamiliar with this framework, specifying the departments involved would enhance clarity and accessibility. Response 1: There is a paragraph about the UDRH initiative on page 2. However, we have expanded this paragraph to read: University Departments of Rural Health (UDRH) are academic centres based in regional, rural and remote (RRR) locations across Australia, and are funded by the Australian Government Department of Health, Disability and Ageing through the Rural Health Multidisciplinary Training Program (RHMT) [17]. The Australian Government has invested in UDRHs for over 25 years as part of their solution to addressing the health workforce shortage that occurs in rural Australia. UDRHs facilitate RRR training and placements for nursing and allied health students, support the continuing professional development of rural health professionals, and undertake research on rural and remote health issues and so are uniquely positioned to help build a culturally safe rural health workforce [18]. UDRHs generally facilitate cultural awareness opportunities for students on placement and may partner with local Indigenous communities, provide mentoring to Indigenous health students, support the local Indigenous health workforce and undertake locally relevant research. Research focus areas include rural and remote health workforce development and improving the health and wellbeing of Indigenous Australians. The Australian Rural Health and Education Network (ARHEN) is the national association for the 19 UDRHs, providing leadership and strategic direction, as well as facilitating UDRHs to share information and collaborate to improve the health and well-being of rural communities [17]. Comment 2: Geographic Representation I was surprised to see that no studies from South Australian locations were included in the review. Could the authors clarify whether this was due to a lack of submissions from that region, or if there were other reasons for their exclusion? Response 2: South Australian articles were represented in the “Research conducted across multiple states” category. All the articles with South Australian locations were done in partnership with other states. There were no workforce articles that came just from South Australia. This may be because Flinders UDRH operates from their Adelaide campus and sites in the Northern Territory (e.g. Alice Springs and Darwin) so a lot of their research occurred across multiple states. The ARHEN UDRH publication database was originally contracted to the University of South Australia to establish so their publications were certainly captured. Moreover, it is important to acknowledge that the extent to which UDRHs have interest and published research related to Aboriginal health and workforce is very uneven – reflecting the staffing and context of each individual UDRH. Comment 3: Scope of Literature Search The decision to limit the review to articles submitted to ARHEN warrants further explanation. Was this a methodological choice based on scope, or were there constraints that prevented a broader search? Clarifying this would strengthen the transparency of the review process. Additionally, the literature search was limited to 2010-2021. Even a quick Google Scholar search shows several articles that could be included in this search. If at all possible, I would encourage the authors to extend their search to include more recent articles. Response 3: The decision to limit the review to articles submitted to ARHEN was a methodological choice as the aim of this narrative review was to examine the contributions of UDRHs to health workforce issues through analysis of their publications focused on Indigenous health. We have attempted to make this clearer by inserting the following into the Introduction: This study was conducted as part of a broader investigation to identify and describe the contributions of UDRHs to Indigenous issues through analysis of the ARHEN Endnote library during the period 2010–2021. The previous study explored the number and type of publications with a focus on Indigenous health that were published by the UDRHs between 2010 and 2021 [19], and as foreshadowed in that article, there was an intention to explore in more detail some of the research categories. Interestingly, despite the focus of UDRHs on building the rural health workforce and supporting Indigenous health, the number of articles (45/493) which specifically addressed workforce issues related to Indigenous health was surprisingly small and we called for further attention in this area [19]. The aim of this review is to examine the workforce articles with a focus on Indigenous health, as identified in our previous study [19], with the intention of identifying the contributions of UDRHs to key learnings relevant to improving Indigenous health and gaps in the research. The literature search was limited to 2010-2021 to be consistent with the first publication. It should be noted that the analysis for this review began in 2023, at which stage 2021 was the most recent full year available for analysis in the ARHEN database. However, as this was a collaborative project, working with colleagues across multiple UDRHs and included building capacity in less experienced researchers, this project took longer than expected. We acknowledge that more recent publications may now be available and have noted this as a limitation in the manuscript. Comment 4: Terminology: “First Nations” vs. “Indigenous” I recommend that the authors consider using the term “First Nations” in place of “Indigenous,” as this is increasingly recognized as the preferred terminology in Australia. A brief note on terminology choices would be helpful for readers. Response 4: Terminology choice is a contested area and there is a diversity of opinions on preferred terminology. We have used the term “Indigenous Australians” in this manuscript to be consistent with the first publication, a decision made at that time, and which was agreed to by the Aboriginal and Torres strait Islander Staff Alliance. However, we have expanded our note on terminology to read: (Hereafter the term ‘Indigenous Australians’ is respectfully used to refer to Australia’s Aboriginal and Torres Strait Islander peoples, while acknowledging the diversity of cultures and experiences of Australia’s First Peoples) Comment 5: Self-Citation Transparency Of the 46 papers reviewed, 23 include at least one author from the current manuscript. While self-citation is not inherently problematic in a review article, I encourage the authors to be more transparent about this overlap. For example: On page 2, in the final paragraph, please clarify that the Thompson article was authored by the majority of the current team. On page 3, revise “was recommended” to “we recommended” to reflect authorship. Consider including a brief statement acknowledging the proportion of self-authored articles and describing any efforts made to identify and include work from other research teams to mitigate potential bias. Response 5: Page 2: We have reworded the final paragraph of the Introduction to make it clearer that the Thompson et al article was authored by the majority of the current team: This study was conducted as part of a broader investigation to identify and describe the contributions of UDRHs to Indigenous issues through analysis of the ARHEN Endnote library during the period 2010–2021. The previous study explored the number and type of publications with a focus on Indigenous health that were published by the UDRHs between 2010 and 2021 [19], and as foreshadowed in that article, there was an intention to explore in more detail some of the research categories. Interestingly, despite the focus of UDRHs on building the rural health workforce and supporting Indigenous health, the number of articles (45/493) which specifically addressed workforce issues related to Indigenous health was surprisingly small and we called for further attention in this area [19]. The aim of this review is to examine the workforce articles with a focus on Indigenous health, as identified in our previous study [19], with the intention of identifying the contributions of UDRHs to key learnings relevant to improving Indigenous health and gaps in the research. Page 3: We have changed the sentence to read: We recommended exploration in greater detail of the 45 research publications on workforce [19] which are the subject of this narrative review. Comment 6: Authorship and Indigeneity On page 13, the authors note the difficulty in reliably coding for Indigenous authorship. However, the Indigeneity of the current authors is not made transparent. A brief statement on this would contribute to the integrity and reflexivity of the work. Response 6: We have added the following on page 3: The research team consisted of two Indigenous researchers; AM, a Mununjahli and Minjungbal woman with a background in education and training and CG, a Wajarri, Badimaya and Wilunyu woman and experienced researcher. The non-Indigenous members of the research team have extensive experience in Indigenous health research, cultural education and training, and rural health. Comment 7: Minor Correction On page 7, please correct the spelling of “Campinha-Bacote.” Response 7: Corrected. Thank you for picking up this mistake.
Reviewer 2 Report
Comments and Suggestions for Authors
clearly define "cultural safety."
Sample: 46 articles identified across the 12 years, 19 focused on developing the understanding and cultural safety skills of university students studying in a health field including campus-based
Indigenous health education and support for students undertaking rural clinical placements. Twelve articles investigated cultural safety skills and recruitment and retention. Are these 46 articles representative? Are there other relevant articles that do not explicitly use the term "cultural safety?"
Document Aboriginal and Torres Strait Islanders disparities in terms of health care. Provide concrete examples of fear and mistrust; give specific instances of racism. Racism in Australia takes different forms than in Europe and North America. Address this.
what would the sample look like if did not have at least one UDRH author?
Only one article talked about the negative impact of racism. Do other articles mention racism?

Author Response
Comment 1: clearly define "cultural safety." Response 1: We have added the following definition of cultural safety in the Introduction on page 2: The National Aboriginal and Torres Strait Islander Health Plan 2021–2031 states that “cultural safety is about how care is provided, rather than what care is provided. It requires practitioners to deliver safe, accessible and responsive health care that is free of racism by recognising and responding to the power imbalance between practitioner and patient; and reflecting on their knowledge, skills, attitudes, practising behaviours, and conscious and unconscious biases” [11]. Comment 2: Sample: 46 articles identified across the 12 years, 19 focused on developing the understanding and cultural safety skills of university students studying in a health field including campus-based Indigenous health education and support for students undertaking rural clinical placements. Twelve articles investigated cultural safety skills and recruitment and retention. Are these 46 articles representative? Are there other relevant articles that do not explicitly use the term "cultural safety?" Response 2: The aim of this review was explicitly to examine the contributions of UDRHs to health workforce issues through analysis of their publications focused on Indigenous health, to this end the 46 articles are representative of the research and findings conducted by the 19 UDRHs. However, the findings of this review support findings from previous Australian health workforce reviews (e.g. Gwynne and Lincoln 2016 and Walsh et al 2020). “Cultural safety” was not a search term that we used. We identified the articles firstly through searching the ARHEN Endnote database for Indigenous health articles, and then reading the articles and coding them, to identify the workforce articles. The term “cultural safety” was a descriptor used by us as to describe what the articles were trying to achieve – even though some articles may not have used this term. However, cultural safety emerges as a key issue for Indigenous patients and for Indigenous providers of health care. Comment 3: Document Aboriginal and Torres Strait Islanders disparities in terms of health care. Provide concrete examples of fear and mistrust; give specific instances of racism. Racism in Australia takes different forms than in Europe and North America. Address this. Response 3: We appreciate your suggestion. While we are mindful of not framing the opening paragraph of the manuscript in a deficit-focused way, we have expanded the Introduction to read: Aboriginal and Torres Strait Islander peoples are the original inhabitants of Australia and experience large, persistent health disparities, with shorter life expectancy and higher rates of chronic diseases compared to non-Indigenous Australians [1]. We have provided some examples of fear, mistrust and racism; however, this review does not focus specifically on racism and racism was not included in our search terms; hence, a discussion of the various forms of racism is not appropriate in the Introduction. We have re-written the relevant sentence to read: Barriers to healthcare include fear or mistrust of mainstream healthcare (due to past negative experiences or fear of negative government interventions) and poor communication, disrespect and racism from health professionals, with experiences of racial bias affecting patients’ willingness to engage with care and leading to misdiagnosis and poorer outcomes [3-5]. Comment 4: what would the sample look like if did not have at least one UDRH author? Response 4: The aim of this review was explicitly to examine the contributions of UDRHs to health workforce issues through analysis of their publications focused on Indigenous health. We have clarified this in the Introduction: The aim of this review is to examine the workforce articles with a focus on Indigenous health, as identified in our previous study [19], with the intention of identifying the contributions of UDRHs to key learnings relevant to improving Indigenous health and gaps in the research. Comment 5: Only one article talked about the negative impact of racism. Do other articles mention racism? Response 5: Note that we did not do searches of these articles on the term racism – since that was not the purpose of our review. Many articles do make mention of the many adverse impacts arising as part of the legacy of colonisation – racism is one of these. Only one article in the 12 health workforce publications mentioned racism, however three of the 19 student workforce publications talk about racism as part of education. This is indicated in Table 3 and described in section 3.2.1 Campus-Based Education.Reviewer 3 Report
Comments and Suggestions for Authors
Thank you for the opportunity to review this paper. Several queries need to be addressed before the reviewer can accept this paper.
- In the methods, the inclusion/exclusion criteria for selecting the articles are only partially described. Please add a clearly defined table that lists the eligibility criteria and explains why some articles were reclassified or excluded during the screening process.
- A narrative review is weaker than a scoping review. Please explain why authors choose only a narrative review and not, at least, a scoping review?
- Please specify the search strategy used in this review.
- In the screening stage, the authors mentioned five names. How does the screening work? Does each of the five reviewers review and screen similar articles, and then another reviewer will resolve the discrepancies? Please explain in detail the process of screening and data extraction, including the tools used for screening and extraction
- in Figure 1. The numbers do not match between steps. From 493 to 45, how many were excluded and for what reason? Please mention it in the flow diagram.
- Why systematic review included? In a systematic review, articles are typically excluded, and only the individual articles within the systematic review are included. Please explain.
- What kind of guidelines did the authors follow for this article? Is it Prisma?
- Why did the authors only use articles from the ARHEN database? Why not include PubMed, Scopus, etc.?
Author Response
Comment 1: In the methods, the inclusion/exclusion criteria for selecting the articles are only partially described. Please add a clearly defined table that lists the eligibility criteria and explains why some articles were reclassified or excluded during the screening process. Response 1: We have added a table (Table 1) that lists inclusion and exclusion criteria for this review. We have also attempted to clarify why some articles were reclassified or excluded as follows: To ensure that no relevant articles were missed, the original set of 493 publications from the previous study were re-examined by the research team and six additional articles were identified and included, bringing the total for this study to 51. In the original breakdown, systematic reviews were reported separately to research articles; however, as the aim of this review was to consider UDRH research contributions three workforce reviews were deemed relevant and included. In addition, three articles originally categorised as “health services research” or “epidemiology” were identified as relevant to workforce issues and included. However, during the initial screening five articles were considered not sufficiently relevant to workforce and were excluded. Article inclusion and exclusion criteria are shown (Table 1). Results for each stage of the identification and screening processes are shown (Fig. 1). Comment 2: A narrative review is weaker than a scoping review. Please explain why authors choose only a narrative review and not, at least, a scoping review? Response 2: The aim of this review was explicitly to examine the contributions of University Departments of Rural Health (UDRHs) to health workforce issues through analysis of their publications focused on Indigenous health. Therefore, a narrative review was appropriate as we wished to synthesize and summarise the findings of UDRHs in the health workforce space. We have clarified the aim of this review in the Introduction: The aim of this review is to examine the workforce articles with a focus on Indigenous health, as identified in our previous study [19], with the intention of identifying the contributions of UDRHs to key learnings relevant to improving Indigenous health and gaps in the research. Comment 3: Please specify the search strategy used in this review. Response 3: Details of the original search strategy were outlined in our previous study. The articles analysed in this review were identified in a previous study. We have edited the Identification and Screening section to make that clear: This narrative review analysed Indigenous health workforce papers that were identified and classified as workforce papers in our previous study [19]. Articles that specifically examined Indigenous issues related to the health workforce published between 2010 and 2021, and with at least one UDRH-affiliated author were included. Our previous study identified 493 UDRH publications in Indigenous health sourced from the ARHEN Endnote database, and coded articles into the following broad categories: health services research; epidemiology, Indigenous culture and needs for health and wellbeing; workforce issues; and other. Details of the search strategy and screening to identify the Indigenous health research articles can be found in our previous publication [19]. The previous study identified 45 research publications on workforce and recommended exploring those publications in greater detail [19] which therefore form the subject of this narrative review. Comment 4: In the screening stage, the authors mentioned five names. How does the screening work? Does each of the five reviewers review and screen similar articles, and then another reviewer will resolve the discrepancies? Please explain in detail the process of screening and data extraction, including the tools used for screening and extraction Response 4: We have clarified the screening and extraction stage as follows: Five authors (EVT, LH, HH, AM and SCT) were each allocated a subset of the 46 papers to screen and review, with all authors reviewing approximately nine articles. Each author extracted information on the key findings and recommendations of their assigned articles as relating to the student workforce, health workforce and Indigenous Australian health workforce using a data extraction template created in Microsoft Excel. The extracted information was cross-checked by at least one additional author to ensure accuracy. Any discrepancies were resolved through discussion within the wider research team. Comment 5: in Figure 1. The numbers do not match between steps. From 493 to 45, how many were excluded and for what reason? Please mention it in the flow diagram. Response 5: The reasons the numbers do not match between steps 1 and 2 in the flow diagram is because those steps were representing the initial analysis reported in the previous study. It was the previous study which identified 45 research articles relating to workforce from the original 493 Indigenous articles. However, we have modified the flow diagram to indicate the 448 articles that identified in the previous study as not relating to workforce. Comment 6: Why systematic review included? In a systematic review, articles are typically excluded, and only the individual articles within the systematic review are included. Please explain. Response 6: The aim of this narrative review was to examine the research outputs of UDRHs with regards to health workforce issues through analysis of their publications focused on Indigenous health. Therefore, it was appropriate to include systematic reviews conducted by UDRHs as they were part of their body of research outputs. We have clarified this in the inclusion criteria and when describing the screening process. Comment 7: What kind of guidelines did the authors follow for this article? Is it Prisma? Response 7: Our manuscript reports on a narrative review not a systematic review, so many of the elements on the PRISMA checklist are not applicable. We followed narrative review methodology in a systematic way, as has been outlined in our flow diagram. Comment 8: Why did the authors only use articles from the ARHEN database? Why not include PubMed, Scopus, etc.? Response 8: The aim of this review was explicitly to examine the contributions of UDRHs to health workforce issues through analysis of their publications focused on Indigenous health. Therefore, we only searched the ARHEN database, which is a central repository of UDRH research. We have clarified this in the Introduction: In 2019 ARHEN established an Endnote database to serve as a central repository of UDRH health, training and workforce research from 2010 onwards. This study was conducted as part of a broader investigation to identify and describe the contributions of UDRHs to Indigenous issues through analysis of the ARHEN Endnote database during the period 2010–2021. The aim of this review is to examine the workforce articles with a focus on Indigenous health, as identified in our previous study [19], with the intention of identifying the contributions of UDRHs to key learnings relevant to improving Indigenous health and gaps in the research.Round 2
Reviewer 1 Report
Comments and Suggestions for Authors
Thank you for responding to my prior suggestions for the improvement of the article. These changes have improved the manuscript and addressed the issues. I wish you all the best with your important work!